# Comparison of vaccination efficacy using live or ultraviolet-inactivated influenza viruses introduced by different routes in a mouse model

**Kyeongbin Baek**[1☉], **Sony Maharjan**[2☉], **Madhav Akauliya**[2☉], **Bikash Thapa**[3],
**Dongbum Kim**[2], **Jinsoo Kim**[1], **Minyoung Kim**[1], **Mijeong Kang**[1], **Suyeon Kim**[1], **Joon-Yong Bae**[4], **Keun-Wook Lee**[3], **Man-Seong Park**[4], **Younghee Lee**[5], **Hyung-Joo Kwon**[1,2]*

1 Department of Microbiology, College of Medicine, Hallym University, Chuncheon, Republic of Korea,
2 Institute of Medical Science, College of Medicine, Hallym University, Chuncheon, Republic of Korea,
3 Department of Biomedical Science, Hallym University, Chuncheon, Republic of Korea, 4 Department of Microbiology, College of Medicine, and the Institute for Viral Diseases, Korea University, Seoul, Republic of Korea, 5 Department of Biochemistry, College of Natural Sciences, Chungbuk National University, Cheongju, Republic of Korea

☉ These authors contributed equally to this work.
* hjookwon@hallym.ac.kr

**Data Availability Statement:** All relevant data are within the paper and its Supporting information files.

## Abstract

Influenza is a major cause of highly contagious respiratory illness resulting in high mortality and morbidity worldwide. Annual vaccination is an effective way to prevent infection and complication from constantly mutating influenza strains. Vaccination utilizes preemptive inoculation with live virus, live attenuated virus, inactivated virus, or virus segments for optimal immune activation. The route of administration also affects the efficacy of the vaccination. Here, we evaluated the effects of inoculation with ultraviolet (UV)-inactivated or live influenza A virus strains and compared their effectiveness and cross protection when intraperitoneal and intramuscular routes of administration were used in mice. Intramuscular or intraperitoneal inoculation with UV-inactivated Influenza A/WSN/1933 provided some protection against intranasal challenge with a lethal dose of live Influenza A/WSN/1933 but only when a high dose of the virus was used in the inoculation. By contrast, inoculation with a low dose of live virus via either route provided complete protection against the same intranasal challenge. Intraperitoneal inoculation with live or UV-inactivated Influenza A/Philippines/2/1982 and intramuscular inoculation with UV-inactivated Influenza A/Philippines/2/1982 failed to produce cross-reactive antibodies against Influenza A/WSN/1933. Intramuscular inoculation with live Influenza A/Philippines/2/1982 induced small amounts of cross-reactive antibodies but could not suppress the cytokine storm produced upon intranasal challenge with Influenza A/WSN/1993. None of the tested inoculation conditions provided observable cross protection against intranasal challenge with a different influenza strain. Taken together, vaccination efficacy was affected by the state and dose of the vaccine virus and the route of administration. These results provide practical data for the development of effective vaccines against influenza virus.

**Funding:** This research was supported by grants from the National Research Foundation (NRF-2020R1A2B5B02001806) funded by the Ministry of Science and ICT in South Korea. The funders had no role in study design, data collection and analysis, decision to publish, or preparation of the manuscript.

**Competing interests:** The authors declare that no competing interests exist.

## Introduction

Influenza is an acute infection of the respiratory tract mainly caused by the influenza A and B viruses. Since the 1918 influenza outbreak, the emergence of new influenza viruses has caused recurrent pandemic and seasonal epidemic outbreaks resulting in substantial morbidity and mortality worldwide [1, 2]. The fragmented nature of the influenza virus genome enables genetic reassortment and the constant generation of genetically and phenotypically distinct variants. This persistent variation presents a major challenge to the development of influenza vaccines [3]. Over the years, there have been many efforts to develop a universal influenza vaccine, but enduring and broad protective immunity is currently still out of reach. Understanding the mechanisms of cross-reactivity and the immune responses elicited by influenza infection and vaccination is vital to generate more effective vaccines [4].

Antibodies against the head of the influenza hemagglutinin (HA) protein have neutralizing activity against multiple influenza subtypes [5–7]. Neutralizing antibodies induced by infection with the pandemic H1N1 2009 strain that bound to the stem and head regions of HA were largely cross-reactive against other influenza strains [8, 9]. Vaccination with live seasonal Flu-Mist vaccine enhances cross-protective T cell immunity against Influenza H1N1 CA04 by inducing $CD4^+$ cells but not $CD8^+$ T cells [10]. Notably, a single-dose intranasal administration of γ-A/PR8[H1N1] was reported to induce cross-protective immunity against Influenza H5N1 and other heterotypic infections, which was mediated by memory cytotoxic T cells [11]. Furthermore, it was reported that vaccination with the seasonal Influenza A/H3N2 virus induced protection against Influenza H5N1 and pH1N1 viruses, and this heterosubtypic immunity was also mediated by a strong T cell response [12, 13].

We previously demonstrated that intraperitoneal inoculation of mice with the Influenza H1N1 strain A/WSN/1933 (WSN) induced cross-reactive antibodies that facilitated heterosubtypic immunity to Influenza H3N2 strain A/Hongkong/4801/2014 [14, 15]. Furthermore, we found that intraperitoneal inoculation with live influenza A virus altered immune cell populations at an early stage, resulting in depletion of B cells and macrophages along with immense neutrophil infiltration in the peritoneal cavity and bone marrow [15]. Expansion of the $CD8^+$ T cell population in response to intraperitoneal inoculation with live influenza A virus likely played a role in cell-mediated protective immunity.

Globally, various types of influenza vaccines have been used, including vaccines based on live attenuated viruses and whole inactivated viruses [16]. In addition, the route and site of immunization profoundly affect vaccine efficacy [17]. We evaluated the effects of vaccination using UV-inactivated virus or live virus administered at different doses intraperitoneally or intramuscularly. We also examined the cross protection conferred by different modes of vaccination.

## Material and methods

### Cell line and viruses

Madin-Darby canine kidney (MDCK) cells obtained from the American Type Culture Collection (ATCC, Manassas, VA, USA) were grown in minimum essential medium (MEM; Thermo Fisher Scientific, Waltham, MA, USA) with 10% fetal bovine serum (FBS; Thermo Fisher Scientific), 100 μg/ml streptomycin, and 100 U/ml penicillin in a 5% $CO_2$ incubator at 37˚C. Influenza A virus subtypes A/WSN/1933(H1N1) (WSN) and A/Philippines/2/1982(H3N2) (H3N2 Php) were used in this study.

### Virus preparations

Single-passage viruses were used to inoculate the allantoic cavity of 9-day old specific pathogen-free (SPF) embryonated chicken eggs. The inoculated eggs were then incubated in a

humidified incubator at 37˚C for 48 h. Allantoic fluid was then collected, centrifuged, and stored at -80˚C prior to use.

MDCK cell monolayers were washed with PBS, infected with influenza A virus at an MOI of 0.01 in MEM containing 1 μg/ml L-tosylamide-2-phenylethyl chloromethyl ketone (TPCK)-treated trypsin (Sigma-Aldrich, Saint Louis, MO, USA), and incubated at 37˚C for 1 h. The inoculum was then removed, and the cells were grown in MEM containing 0.3% BSA for 72 h. Supernatants were then collected and centrifuged at 2,000 rpm for 10 min at 4˚C to remove the cell debris. The presence of amplified influenza A viruses in the collected supernatant was verified by plaque assay. All work related to virus propagation and cell culture in this study was performed in Biosafety level 2 conditions.

## Plaque assay

Plaque assays were performed using MDCK cells as described previously [18]. Briefly, $7 \times 10^5$ cells/well were seeded on six-well plates and incubated at 37˚C for 18 h. The confluent MDCK cell monolayer was then washed with PBS, inoculated with tenfold serial dilutions of virus stocks and lung homogenates, and incubated for 1 h at 37˚C with periodic shaking at 15 min intervals. Unabsorbed virus was then removed, and the cells were overlaid with 2 ml DMEM/F12 medium (Thermo Fisher Scientific) containing 4% BSA, 10 mM HEPES, 2 mM glutamine, 50 mg/ml DEAE dextran, 2.5% sodium bicarbonate, 1 μg/ml TPCK-treated trypsin, 100 μg/ml streptomycin, 100 U/ml penicillin, and 0.6% immunodiffusion-grade agar. After incubation for 72 h at 37˚C in a 5% $CO_2$ incubator, the cells were stained with 0.1% crystal violet, plaques were counted, and the virus titers were calculated.

## Ultraviolet inactivation of influenza A virus

Inactivation of influenza virus was carried out as previously described [19]. Briefly, viruses were irradiated with 254 nm UV light at distance of 5 cm from a UV light source for 15 min. Inactivation of the viruses was confirmed by plaque assays showing no plaque-forming units after the UV exposure.

## Ethics statement

All animal studies were conducted in accordance with the recommendations in the Guide for the Care and Use of Laboratory Animals of the National Veterinary Research & Quarantine Service of Korea. The Institutional Animal Care and Use Committee (IACUC) of Hallym University (Permit Number: Hallym2021-70) approved the animal experiments in this study. Research staff involved in animal care or handling took an education course for the users of experimental animal facility in Laboratory Animal Resources Center of Hallym University. They also took an education course for Biosafety Level 3 at Korea Human Resource Development Institute for Health & Welfare (KOHI). Exposure to 1–2% isoflurane (Pharmaceutical, Seoul, Korea) was used to anesthetize mice for virus infection. To collect lungs and blood by cardiac puncture, mice were anesthetized by intraperitoneal injection of 0.2 ml 2,2,2-tribromoethanol in *tert*-amyl alcohol (Avertin; Sigma-Aldrich). Health and behavior of experimental animals were monitored daily, and the bedding was changed once a week, which ensures a regulatory compliance for the welfare of laboratory animals. Humane endpoints were planned to euthanize the mice by $CO_2$ inhalation in accordance to the approved IACUC protocol when the mice lose 30% of adult body weight or exhibit evidence of debilitation, pain or distress such as a hunched posture, rough haircoat, reduced food consumption, emaciation, inactivity, difficulty ambulating, respiratory problems. Among 510 mice used, 120 mice were found dead before they reached the endpoint criteria. After experiments were terminated, remnant mice

(n = 390) were euthanized by $CO_2$ inhalation, and all efforts were made to minimize animal pain and suffering.

## Mice immunization and infection

Four-week-old female BALB/c (H-2[b]) mice were obtained from Nara Biotech, Inc. (Seoul, Korea) and maintained in environmentally controlled SPF rooms with a 12-h light/dark cycle at 20–25˚C with 40–45% humidity and *ad libitum* access to food and water. All animal experiments involving virus infection were performed under animal biosafety level 2 conditions in the Research Institute of Medical-Bio Convergence of Hallym University in accordance with the recommendation of the Institutional Biosafety Committee of Hallym University. During the experimental period, mice were maintained in an individually ventilated cage (IVC) under a 12-h light/dark cycle at 20–25˚C with 40–45% humidity and *ad libitum* access to food and water. Mice were inoculated intraperitoneally or intramuscularly with live or UV-inactivated WSN or H3N2 Php at a dose of $5×10^6$ pfu or $5 × 10^7$ pfu per mouse. The live influenza virus used here is not a live attenuated vaccine. At 14 days post inoculation (dpi), the mice (n = 10/ group) were challenged intranasally with 10 $LD_{50}$ of live WSN virus as described previously [19]. The mice were monitored daily for clinical signs and body weight for up to 10 days.

## Sample collection

To examine virus-specific antibody production and cytokine production, blood samples (n = 5/group) were collected via retro-orbital bleeding 14 days after intraperitoneal or intramuscular inoculation and via cardiac puncture 5 days after intranasal challenge. Serum samples were prepared and stored at -80˚C. The mice were sacrificed 5 days after intranasal challenge, and the lungs were removed for analysis. Lungs (n = 5/group) were weighed and homogenized using Tissue Lyser II (Qiagen, Hilden, Germany), and virus titers were determined by plaque assay. For histopathologic examination, lungs (n = 5/group) were fixed in 4% paraformaldehyde, embedded in paraffin, and sectioned at 5 μm thickness. The specimens were then stained with Gill's Hematoxylin V (Muto Pure Chemicals, Tokyo, Japan) and Eosin Y solution (Sigma-Aldrich).

## Measurement of cytokines in mouse lungs and sera

Cytokines in sera and lung homogenates were measured using a Cytometric Bead Array (CBA) Mouse Th1/Th2/Th17 Cytokine Kit (BD Biosciences, San Jose, CA, USA. Catalog No: 560485). The CBA experiments were performed in accordance with the manufacturer's instructions as described previously [14]. The CBA kit contained a mixture of seven different capture beads with distinct fluorescent intensities coated with antibodies specific for IL-2, IL-4, IL-6, IL-10, IL-17A, TNF, and IFN-γ. The lung homogenates and sera were analyzed using FACSCalibur (BD Bioscience), and the levels of cytokines were quantified using the LEGEN-Dplex™ software, version 7.0 (BioLegend, San Diego, CA, USA).

## ELISA

Ninety-six-well immunoplates (Nunc™, Roskilde, Denmark) were coated with live WSN or H3N2 Php in carbonate buffer (pH 9.6) and incubated overnight at 4˚C. After blocking with 1% BSA, threefold dilutions of sera in PBST were added to the plates, which were then incubated for 2 h at room temperature. The plates were then washed three times with PBST, and horseradish peroxidase (HRP)-conjugated goat anti-mouse IgG antibody (1:500 dilution; Catalogue No: 5300–05, Southern Biotechnology Associates, Inc., Birmingham, AL, USA) was

added. After 1 h incubation at room temperature, the plates were washed three times with PBST and developed colorimetrically using TMB (3,3',5,5'-tetramethylbenzidine) substrate solution (Kirkegaard and Perry Laboratories, Gaithersburg, MD, USA). The absorbance at 450 nm was then measured using a SpectraMax 250 microplate reader (Molecular Devices, Sunnyvale, CA, USA).

## Hemagglutination inhibition (HI) assay

Ninety-six-well V-bottom plates (Costar, Corning, NY, USA) were used for the HI assay as described previously [15]. Receptor-destroying enzyme-treated serum samples were serially diluted two-fold with PBS and then incubated with an equal volume of 4 hemagglutination units (4HA) of WSN or H3N2 Php for 30 min. After incubation, an equal volume of 0.5% chicken red blood cells (Innovative Research, Novi, MI, USA) were added to the wells and incubated for 30 min at room temperature, and HI titers were measured.

## Statistical analysis

Results are shown as the mean ± standard deviation. Differences between two samples were evaluated using Student's t-test with P < 0.05 as the threshold for statistical significance.

# Results

## Intraperitoneal immunization with a high dose of UV-inactivated Influenza A/WSN/1933 induced a prophylactic effect against Influenza A/WSN/1933 challenge

To determine whether immunization with UV-inactivated WSN (UV-WSN) has a protective effect against subsequent challenge with live WSN, mice were intraperitoneally inoculated with a low ($5 \times 10^6$ pfu) or high ($5 \times 10^7$ pfu) dose of UV-WSN without adjuvants and challenged intranasally 14 days later with live WSN. The mice were then monitored for a further 10 days. The low-dose inoculation did not confer any protection in terms of reduced weight loss or increased survival after the intranasal challenge with the live virus, whereas the high-dose inoculation improved survival and reduced weight loss after the intranasal challenge (Fig 1A and 1B). We checked the production of WSN-specific antibody (IgG) in the sera of the mice by ELISA and found that the high-dose intraperitoneal inoculation produced a higher level of WSN-specific IgG than the low-dose intraperitoneal inoculation (Fig 1C). These results showed that the high-dose inoculation with UV-WSN protected mice against a subsequent lethal dose of live WSN.

## Intraperitoneal inoculation with live Influenza A/WSN/1933, but not live Influenza A/Philippines/2/1982, provided protection against Influenza A/WSN/1933 challenge

Cross protection against different strains of influenza is desirable for effective vaccines. To study the cross-protective effect of inoculation with different influenza strains, we injected mice intraperitoneally with PBS or a low dose ($5 \times 10^6$ pfu/mouse) of either live WSN or live H3N2 Php and challenged the mice 14 days later with intranasal administration of a lethal dose (10 $LD_{50}$) of live WSN. In contrast to low-dose inoculation with UV-WSN, low-dose inoculation with live WSN virus provided protection against subsequent intranasal challenge with a lethal dose of the same virus, as shown by a 100% survival rate and no observable weight loss (Fig 2A and 2B), no obvious lung pathology (Fig 2C and 2E), and evidence of viral clearance in plaque assays of lung homogenates (Fig 2D). In contrast, inoculation with live H3N2

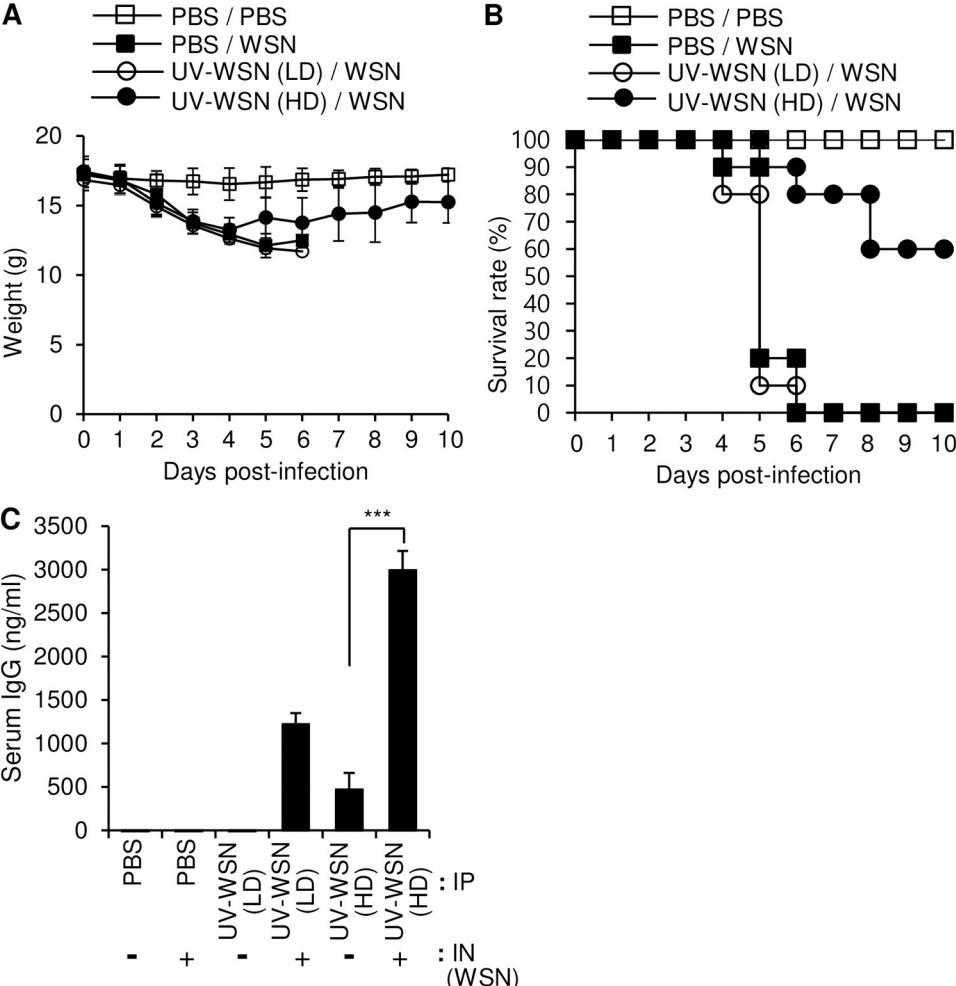

**Fig 1. Intraperitoneal inoculation with a high dose of UV-inactivated Influenza A/WSN/1933 provided partial protection against lethal challenge with Influenza A/WSN/1933.** BALB/c mice were inoculated intraperitoneally with $5 \times 10^6$ pfu (low dose, LD) or $5 \times 10^7$ pfu (high dose, HD) of UV-inactivated Influenza A/WSN/1933 (UV-WSN). At day 14 post inoculation, the mice were challenged intranasally with $10 \ LD_{50}$ of live Influenza A/WSN/1933 (WSN). (A, B) Survival (A) and body weight (B) were evaluated over a 10-day period after the challenge (n = 10/group). (C) Blood samples were collected at day 14 after initial intraperitoneal inoculation or at day 5 after intranasal challenge, and amounts of WSN-specific IgG in sera were determined by ELISA (n = 5/group). $^{***}p < 0.001$.

Php did not confer any protection against weight loss or mortality after intranasal challenge with WSN (Fig 2A and 2B). Morbidity was further confirmed in the mice inoculated with H3N2 Php by an abnormal husky red color, increased viral load, severe alveolar damage, and invasion by inflammatory cells within lung tissues (Fig 2C–2E). The serum levels of WSN-specific IgG 5 days after the intranasal challenge were higher in the mice that were pre-inoculated with WSN than in control mice that were injected with PBS prior to the intranasal challenge (Fig 2F, left panel). By contrast, no WSN-specific IgG was detected in the mice that were pre-inoculated with H3N2 Php (Fig 2F, left panel), although these mice did produce high levels of H3N2 Php-specific IgG (Fig 2F, right panel). When the mice pre-inoculated with H3N2 Php was intranasally challenged with WSN, WSN-specific IgG was produced with a level similar to that of the WSN-inoculated mice. Importantly, WSN challenge alone did not induce detectable production of WSN-specific IgG in the PBS control mice. Therefore, H3N2 Php inoculation is

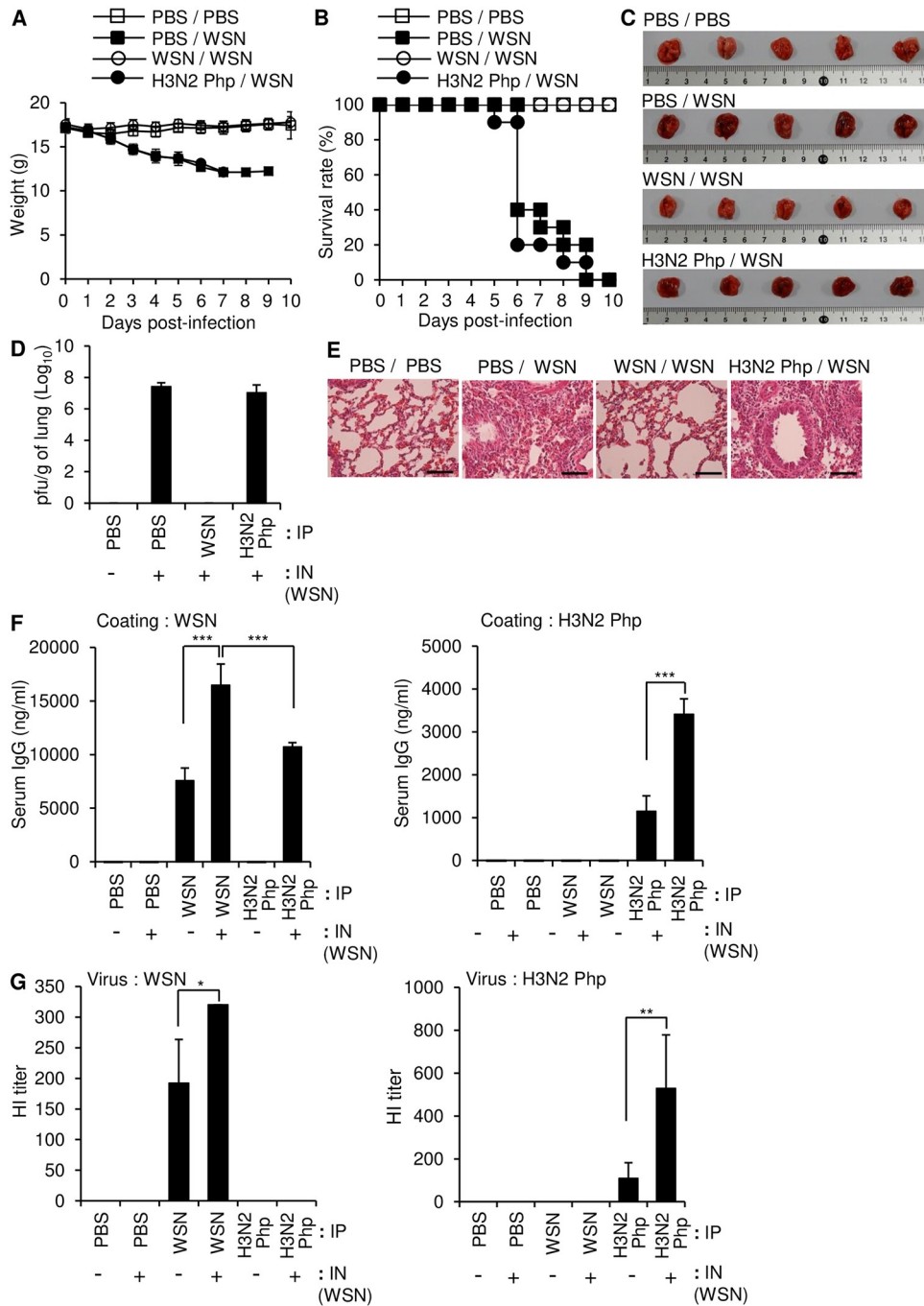

**Fig 2. Intraperitoneal inoculation with a low dose of live Influenza A/WSN/1933 provides complete protection against intranasal challenge with Influenza A/WSN/1933.** BALB/c mice were inoculated intraperitoneally with $5 \times 10^6$ pfu of live Influenza A/WSN/1933 (WSN) or live Influenza A/Philippines/2/1982 (H3N2 Php). At day 14 post inoculation, the mice were challenged intranasally with 10 $LD_{50}$ of WSN. (A, B) Survival (A) and body weight (B) were evaluated over a 10-day period after the intranasal challenge (n = 10/group). (C) Lungs were observed macroscopically 5 days after the intranasal challenge n = 5/group. (D) Viral titers in lung homogenates were determined by plaque assay 5 days after the intranasal challenge. (E) H&E staining of the paraffin-embedded lung sections collected 5 days after the intranasal challenge. Scale bars: 25 μm. (F) The level of WSN-specific total IgG (left) and H3N2 Php-specific total IgG (right) in sera were determined by ELISA (n = 5/group). (G) Hemagglutination inhibition (HI) assay. HI titers of each serum sample were determined with 4 hemagglutination units (4HA) of WSN (left) or H3N2 Php (right). $^*p<0.05$, $^{**}p<0.01$, $^{***}p < 0.001$.

presumed to activate the immune system even though it can't induce WSN-specific antibody production. Production of H3N2 Php-specific IgG was further increased by intranasal challenge with WSN suggesting that WSN can activate production of H3N2 Php-specific IgG possibly via cellular immunity. Importantly, the serum of WSN-inoculated mice inhibited hemagglutination mediated by WSN, but did not show any cross-reactivity to inhibit hemagglutination by H3N2 Php (Fig 2G). Taken together, these results indicate that intraperitoneal inoculation with live H3N2 Php did not induce any detectable WSN-binding antibody or protect against subsequent intranasal exposure to WSN.

## Intramuscular inoculation with UV-inactivated Influenza A/WSN/1933 conferred a small prophylactic effect against Influenza A/WSN/1933 challenge

The most common immunization route in humans is intramuscular injection, which is generally convenient and effective [20]. We therefore investigated the protective effect of intramuscular injections of low ($5 \times 10^6$ pfu/mouse) and high ($5 \times 10^7$ pfu/mouse) doses of UV-WSN or UV-inactivated H3N2 Php (UV-H3N2 Php) against intranasal challenge with a lethal dose of live WSN (10 LD$_{50}$). Intramuscular inoculation with the low dose of either UV-inactivated strain did not confer protective or cross-protective effects against the subsequent intranasal challenge, as all the mice succumbed to infection, exhibited weight loss, and died within 7 days post challenge (Fig 3A and 3B). Macroscopic features, histological features, and viral load within the lungs were similar between the mice with low-dose intramuscular inoculations and control mice that were not inoculated prior to intranasal challenge (Fig 3C–3E). In contrast, the high-dose intramuscular inoculation with UV-WSN elicited a moderate prophylactic effect, as shown by less weight loss and 40% survival after the intranasal challenge (Fig 4A and 4B). Additionally, the lungs of the mice inoculated with the high dose of UV-WSN appeared normal on macroscopic examination, with histological evidence of moderate pathology in lung sections and a slight reduction in viral load within the lungs (Fig 4C–4E). By contrast, high-dose immunization with UV-H3N2 Php did not show any protective effects in comparison with control mice that were not inoculated prior to intranasal challenge (Fig 4B–4E). Notably, the high-dose intramuscular inoculation with UV-WSN did not produce IgG specific for WSN or H3N2 Php in the mice (Fig 4F).

## Intramuscular inoculation with live Influenza A/WSN/1933, but not live Influenza A/Philippines/2/1982, protected against subsequent intranasal challenge with Influenza A/WSN/1933

Because intramuscular inoculation with UV-inactivated virus was only moderately effective to prevent morbidity and mortality after subsequent intranasal exposure to live virus, we tested the protective effects of intramuscular inoculation with low doses of live virus. Mice were intramuscularly inoculated with $5 \times 10^6$ pfu live WSN or live H3N2 Php and then challenged intranasally with a lethal dose (10 LD$_{50}$) of live WSN. Intramuscular inoculation with live WSN conferred complete protection against the subsequent intranasal challenge, as the inoculated mice displayed no weight loss and 100% survival after the challenge (Fig 5A and 5B) and had macroscopic features, viral load, and tissue histology of the lungs that were similar to those in uninfected mice (Fig 5C–5E). In contrast to inoculation with UV-inactivated virus, the inoculation with live WSN induced a high level of WSN-specific serum IgG production, which explains the complete protection against the intranasal challenge (Fig 5F, left panel). In contrast, intramuscular inoculation with live H3N2 Php did not improve weight loss, mortality, gross lung inflammation, lung histology, or viral load after the intranasal challenge in

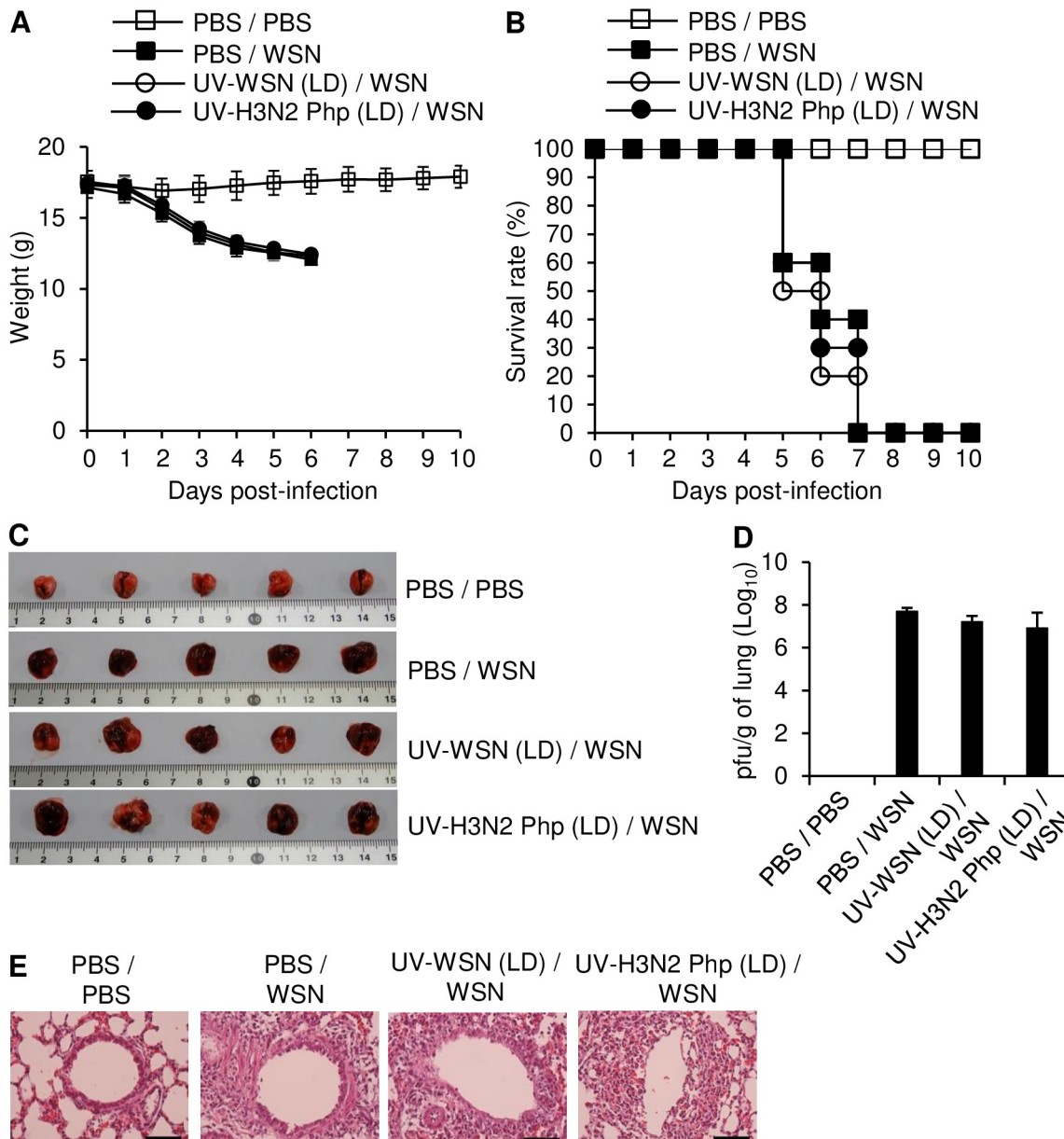

**Fig 3. Intramuscular immunization with a low dose of UV-inactivated virus had no protective effect against intranasal challenge with live virus.** BALB/c mice were inoculated intramuscularly with $5 \times 10^6$ pfu of UV-inactivated Influenza A/WSN/1933 (UV-WSN) or UV-inactivated Influenza A/Philippines/2/1982 (UV-H3N2 Php). At day 14 after inoculation, the mice were challenged intranasally with 10 $LD_{50}$ of live Influenza A/WSN/1933 (WSN). (A, B) Survival (A) and body weight (B) were evaluated over a 10-day period after the intranasal challenge (n = 10/group). (C–E) Lung tissues and blood were collected 5 days after the intranasal challenge (n = 5/group). (C) Macroscopic features of the lung were examined. (D) Viral titers in lung homogenates were determined by plaque assay. (E) Paraffin-embedded sections of lung tissue were stained with H&E. Scale bars: 25 μm.

comparison with control mice that were inoculated with PBS prior to the intranasal challenge (Fig 5A–5E).

Intramuscular inoculation with live H3N2 Php induced production of WSN-specific IgG in the serum; however, the amount was smaller than that produced by intramuscular inoculation with live WSN (Fig 5F, left panel). Production of the WSN-specific antibody was further

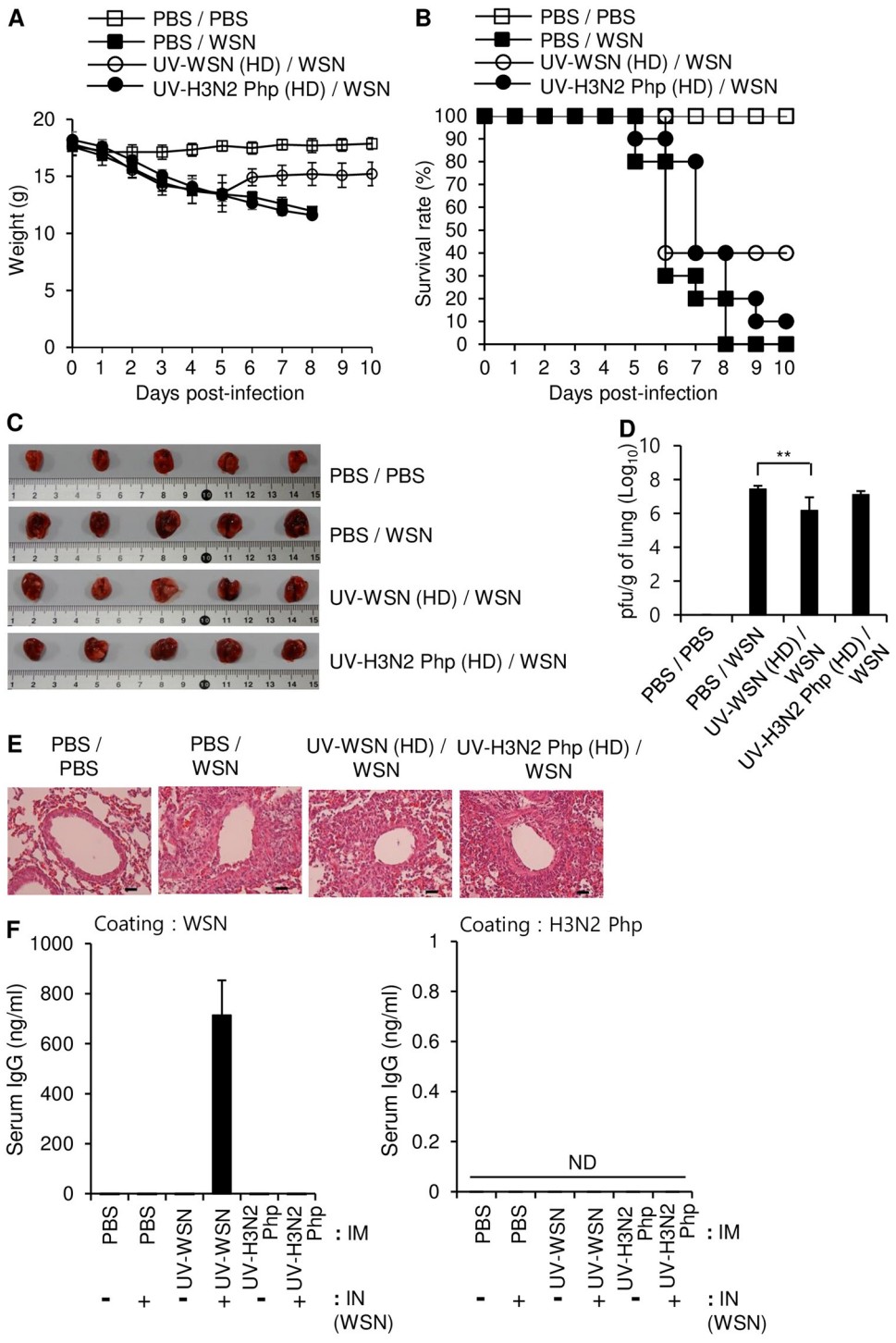

**Fig 4. Intramuscular inoculation with a high dose of UV-inactivated virus provided partial protection against challenge with live virus.** BALB/c mice were intramuscularly inoculated with $5 \times 10^7$ pfu of UV-inactivated Influenza A/WSN/1933 (UV-WSN) or UV-inactivated Influenza A/Philippines/2/1982 (UV-H3N2 Php). After 14 days, the mice were challenged intranasally with 10 $LD_{50}$ of live Influenza A/WSN/1933 (WSN). (A, B) Survival (A) and body weight (B) were measured for 10 days after the intranasal challenge (n = 10/group). (C–F) Lungs and sera were collected 5 days after the intranasal challenge (n = 5/group). (C) Lungs were evaluated macroscopically. (D) The viral load in the lungs was measured by plaque assay. (E) Paraffin-embedded lung sections were stained with H&E. Scale bars: 25 μm. (F) The amounts of WSN-specific (left) and H3N2 Php-specific (right) total IgG in the sera were determined by ELISA (n = 5/group). ND, not detected. $^{**}p < 0.01$.

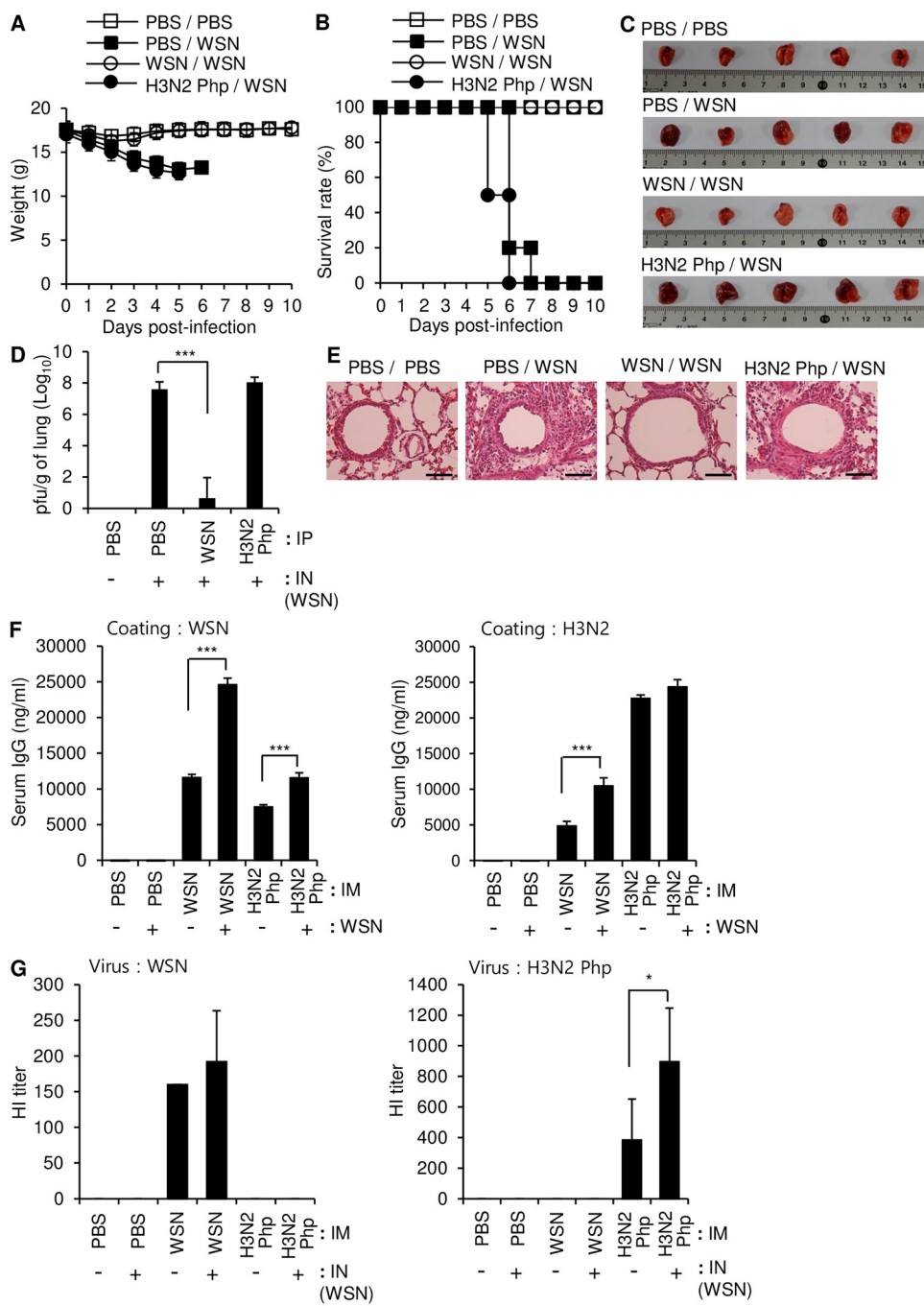

**Fig 5. Intramuscular inoculation with live virus protected against intranasal challenge with the same virus.** BALB/c mice were intramuscularly inoculated with $5 \times 10^6$ pfu of live Influenza A/WSN/1933 (WSN) or live Influenza A/Philippines/2/1982 (H3N2 Php). After 14 days, the mice were challenged intranasally with 10 $LD_{50}$ of live WSN. (A, B) Survival (A) and body weight (B) were measured for 10 days after the intranasal challenge (n = 10/group). (C–F) Lungs and blood were collected 5 days after the intranasal challenge (n = 5/group). (C) The lungs were examined macroscopically. (D) Viral titers in the lungs were measured by plaque assay. (E) H&E staining was performed on formalin-fixed, paraffin-embedded tissue sections of the lungs. Scale bars: 25 μm. (F) The amounts of WSN-specific (left) and H3N2 Php-specific (right) total IgG in sera were determined by ELISA (n = 5/group). (G) Hemagglutination inhibition (HI) assay. HI titers of each serum sample were determined with 4 hemagglutination units (4HA) of WSN (left) or H3N2 Php (right). $^*p<0.05$, $^{***}p<0.001$.

increased in the H3N2 Php-inoculated mice by the intranasal challenge with live WSN; however, the highest measured antibody level was comparable to that in control mice that were inoculated intramuscularly with live WSN without further challenge, which explains the absence of cross protection from the intramuscular inoculation. As expected, intramuscular inoculation with live H3N2 Php also induced production of H3N2 Php-specific IgG; however, the intranasal challenge with live WSN did not significantly enhance the production of that antibody. Conversely, intramuscular inoculation with live WSN induced production of H3N2 Php-specific IgG (Fig 5F, right panel), and the production of that antibody was enhanced by the intranasal challenge with live WSN. The serum of WSN-inoculated mice inhibited hemagglutination mediated by WSN but didn't have any inhibitory effect against hemagglutination induced by H3N2 Php (Fig 5G) as shown for intraperitoneal inoculated mice in Fig 2G. Taken together, these results suggest that low-dose intramuscular inoculation with live virus confers complete protection against subsequent infection with the same virus but provides no cross protection against other viral subtypes.

## Cytokine profiles in the sera of mice inoculated intramuscularly with live virus

During influenza infection, the level of cytokine storm determines the severity of the disease [21]. Therefore, we examined the cytokine levels in sera and lung tissues at various time points of immunization and infection. Cytokine levels in the sera and lungs of mice inoculated intramuscularly with live WSN or live H3N2 Php were similar to those in control mice injected with PBS (Fig 6A–6E); only IFN-γ was slightly increased at day 5 in the sera of the mice intramuscularly inoculated with live H3N2 Php (Fig 6B). These results indicate that intramuscular inoculation with live virus did not significantly affect systemic inflammation, which suggests that it would be safe for healthy individuals.

Mice that were inoculated intramuscularly with live WSN and then challenged intranasally 14 days later with the same virus had the same cytokine levels as uninfected control mice (Fig 6C), whereas mice that were pre-inoculated with live H3N2 Php prior to intranasal challenge with live WSN had increased IFN-γ and IL-6 levels in the sera and increased IL-6 levels in the lungs compared with uninfected control mice (Fig 6C and 6F). Mice that were inoculated with PBS prior to the intranasal challenge had INF-γ and IL-6 levels in the lungs that were similar to those in the uninfected control mice (Fig 6F). These results suggest that the reduced morbidity and increased survival observed in mice that were intramuscularly inoculated with live WSN can be attributed to enhanced IgG production and suppression of inflammatory cytokines. Conversely, the absence of cross protection against H3N2 Php in WSN-inoculated mice might be related to the absence of functionally cross-reactive antibodies and failure to suppress inflammatory cytokine levels. Additionally, the cytokine levels in sera were not different than those in uninfected mice when the mice were intranasally challenged with live WSN 14 days after intraperitoneal inoculation with live H3N2 Php (S1 Fig). This suggests that the effect of the vaccine on cytokine regulation depends on vaccination route.

## Discussion

Reemerging influenza virus subtypes cause high morbidity and mortality and represent a perpetual global threat [22]. Vaccination is one of the most cost-effective public health interventions against seasonal and pandemic influenza outbreaks. Currently, the standard vaccines protect against influenza by eliciting a neutralizing-antibody response to the viral HA and neuraminidase proteins, but they are unable to protect against new subtypes [23, 24]. One of the limitations to efficient vaccination arises from the fact that influenza viruses continuously

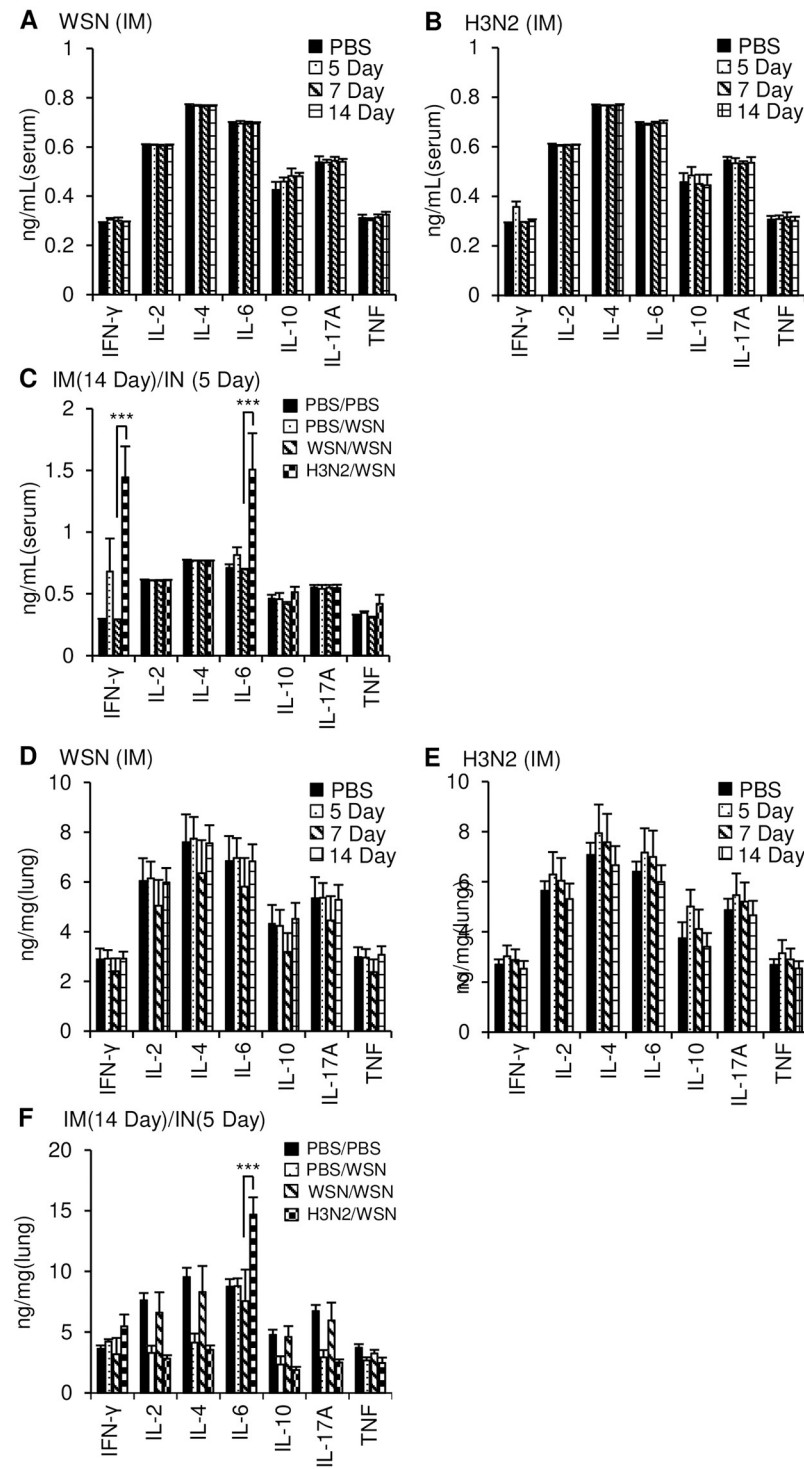

**Fig 6. Cytokine levels in the sera and lung homogenates after intramuscular inoculation with live virus and/or intranasal virus challenge.** BALB/c mice (n = 5) were intramuscularly injected with PBS, $5 \times 10^6$ pfu of live Influenza A/WSN/1933 (WSN) (A, D) or live Influenza A/Philippines/2/1982 (H3N2 Php) (B, E). Blood (A, B) and lungs (D, E) were collected after mice were sacrificed at the indicated time points after inoculation. Sera and lung homogenates were prepared, and levels of cytokines were quantified using a cytokine bead array. (C, F) BALB/c mice (n = 5) were intramuscularly inoculated with PBS, live WSN or live H3N2 Php. After 14 days, the mice were challenged intranasally with PBS or 10 $LD_{50}$ of WSN. Sera (C) and lung homogenates (F) were prepared at day 5 after the challenge, and levels of cytokines were measured by cytokine bead array. $^{***}p < 0.001$.

undergo changes via antigenic drift and shift. Moreover, the traditional method of manufacturing influenza vaccines using eggs is time consuming and expensive [25]. A vaccine that induces cross protection against unpredictable virus strains and can be manufactured promptly and economically is urgently required. Therefore, we examined factors that might be expected to impact on the effectiveness of vaccination. We first examined the abilities of inoculation with live and UV-inactivated influenza viruses to protect against subsequent lethal influenza virus challenge in a mouse model. We then determined cross-protective efficacy using inoculation and challenge with two different influenza strains. In addition, we compared the breadth of protection provided by different routes of immunization.

Previously, we reported that intraperitoneal inoculation of mice with WSN induced cross protection against a lethal intraperitoneal dose of Influenza A/Hongkong/4801/2014, which was likely due to an increase in the CD8$^+$ T cell population and cell-mediated protective immunity in response to the intraperitoneal inoculation [15]. In addition, we found that intraperitoneal inoculation with WSN induced immunity against intranasal exposure to Influenza A/Hongkong/4801/2014 [14].

Live-attenuated vaccines have been associated with highly successful global vaccination campaigns [26]. Both live-attenuated vaccines and inactivated-virus vaccines have been shown to be efficient and safe [27–32]; however, better protection is occasionally observed with live-attenuated vaccines [33–36]. Immunization with del-NS1 live-attenuated vaccine in mice provided protection against pandemic H1N1, H5N1, and H7N9 influenza viruses [37]. Studies in mice and ferrets using Influenza A/Ann Arbor/6/60 cold-adapted (ca) donor strain (H2N2) and Influenza A/Ann Arbor/6/60 ca-based 2009 pandemic H1N1 vaccine demonstrated the protective efficacy of live reassortant virus vaccines against H1N1 and H5N1 viruses [38, 39].

Previous studies in mice showed that intraperitoneal inoculation with live influenza virus confers protection against subsequent intranasal infection [40, 41]. We inoculated mice intraperitoneally with low and high doses of UV-WSN and then challenged them with a lethal intranasal dose of live WSN. The results showed that the high-dose inoculation with UV-inactivated virus provided substantial protection (~40%) against the intranasal exposure, whereas the low-dose inoculation did not confer any protection. Furthermore, we inoculated mice intraperitoneally with a low dose of live WSN or live H3N2 Php and then challenged them intranasally with a lethal dose of live WSN. The inoculation with live WSN conferred significantly more protection than the inoculation with UV-WSN, which is in line with previous findings that live virus is more effective than UV-inactivated virus [15]. In contrast, the intraperitoneal inoculation with live H3N2 Php did not provide any cross protection against intranasal challenge with live WSN.

The vaccine composition and route of administration are important parameters that affect the quality of vaccine response. The vast majority of licensed vaccines are administered via the intramuscular route [42], because conventional vaccination with aluminum-salt adjuvant led to severe adverse reactions when subcutaneous injection was used [20, 43]. Intramuscular injection of influenza vaccines was found to be more immunogenic than subcutaneous injection in elderly adults [44]. We found that intramuscular immunization with a high dose of UV-inactivated virus was only moderately effective to prevent morbidity and mortality due to subsequent intranasal infection. Similar to intraperitoneal inoculation, low-dose intramuscular inoculation did not confer any cross protection against intranasal infection with a different influenza strain, although it provided complete protection against intranasal infection with the same strain. Considering our previous results that intraperitoneal inoculation with WSN provided protection against subsequent challenge with Influenza A/Hongkong/4801/2014 [14, 15], cross-reactivity among different virus strains probably depends on the degree of homology of the viral gene sequences. In this study, we mainly revealed antibody production and

compared outcome of the vaccinated mice after intranasal challenge. Considering that cellular immunity plays an important role in regulating the humoral responses and cellular responses toward the conserved genes of viruses may be critical for effective cross protection against heterologous viruses [4, 24], further investigation on cellular immunity including T cell responses is required.

The cytokine storm produced in response to influenza infection is known to determine the severity of disease [21]. Although higher cytokine levels during primary immunization were previously shown to enhance protection against a secondary challenge with influenza virus in mice [45], failure to lower the cytokine levels over the course of infection can result in severe damage to organs and eventually death [21]. We found that mice inoculated with H3N2 Php had elevated levels of IFN-γ and IL-6 after subsequent challenge with WSN, even at 5 days after the challenge. Hence, an inability to suppress the cytokine storm might have contributed to the lack of cross protection afforded by the initial inoculation. Further studies using various doses and different subtypes of live virus will provide more data on potential cross protection.

In summary, our study provides experimental evidence of the prophylactic effect of intraperitoneal and intramuscular immunizations with UV-inactivated or live influenza virus against subsequent intranasal exposure to live influenza virus. Overall, inoculation with live virus was more protective than inoculation with UV-inactivated virus, and the intramuscular and intraperitoneal routes of administration provided similar levels of protection when live virus was used.

## Supporting information

**S1 Fig. Cytokine levels in the sera after intraperitoneal inoculation with live virus and/or intranasal virus challenge.** BALB/c mice (n = 5) were intraperitoneally inoculated with PBS, live WSN or live H3N2 Php. After 14 days, the mice were challenged intranasally with PBS or 10 $LD_{50}$ of WSN. Sera were prepared at day 5 after the challenge, and levels of cytokines were measured by cytokine bead array.
(TIF)

## Author Contributions

**Conceptualization:** Man-Seong Park, Younghee Lee, Hyung-Joo Kwon.

**Data curation:** Kyeongbin Baek, Sony Maharjan, Madhav Akauliya, Keun-Wook Lee, Man-Seong Park, Younghee Lee, Hyung-Joo Kwon.

**Formal analysis:** Kyeongbin Baek, Sony Maharjan, Madhav Akauliya, Bikash Thapa.

**Funding acquisition:** Hyung-Joo Kwon.

**Methodology:** Kyeongbin Baek, Madhav Akauliya, Bikash Thapa, Dongbum Kim, Jinsoo Kim, Minyoung Kim, Mijeong Kang, Suyeon Kim, Joon-Yong Bae, Keun-Wook Lee.

**Supervision:** Hyung-Joo Kwon.

**Writing – original draft:** Sony Maharjan, Hyung-Joo Kwon.

**Writing – review & editing:** Man-Seong Park, Younghee Lee.

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
