## [Decision Letter · Decision Letter 0]

8 Aug 2022

PONE-D-22-17990Comparison of vaccination efficacy using live or UV-inactivated influenza viruses introduced by different routes in a mouse modelPLOS ONE

Dear Dr. Kwon,

Thank you for submitting your manuscript to PLOS ONE. After careful consideration, we feel that it has merit but does not fully meet PLOS ONE’s publication criteria as it currently stands. Therefore, we invite you to submit a revised version of the manuscript that addresses the points raised during the review process.

 Dear Dr. Hyung-Joo Kwon, Your paper has been reviewed by two experts in the field who state the significant of the studies presented in your paper. However, both reviewers cited a number of weaknesses that need to be adequately addressed. It would be especially important to address the comments by reviewer 2 who has some significant concerns about some technical aspects of the work. Please submit your revised manuscript by Sep 22 2022 11:59PM. If you will need more time than this to complete your revisions, please reply to this message or contact the journal office at plosone@plos.org. Please include the following items when submitting your revised manuscript:A rebuttal letter that responds to each point raised by the academic editor and reviewer(s). You should upload this letter as a separate file labeled 'Response to Reviewers'.A marked-up copy of your manuscript that highlights changes made to the original version. You should upload this as a separate file labeled 'Revised Manuscript with Track Changes'.An unmarked version of your revised paper without tracked changes. You should upload this as a separate file labeled 'Manuscript'.

We look forward to receiving your revised manuscript.

Kind regards,

Juan Carlos de la Torre, Ph.D.

Academic Editor

PLOS ONE

Journal Requirements:

"This research was supported by grants from the National Research Foundation (NRF-2020R1A2B5B02001806) funded by the Ministry of Science and ICT in South Korea."

Reviewers' comments:

Reviewer's Responses to Questions

**Comments to the Author**

1. Is the manuscript technically sound, and do the data support the conclusions?

Reviewer #1: Yes

Reviewer #2: Partly

2. Has the statistical analysis been performed appropriately and rigorously? 

Reviewer #1: Yes

Reviewer #2: Yes

3. Have the authors made all data underlying the findings in their manuscript fully available?

Reviewer #1: Yes

Reviewer #2: Yes

4. Is the manuscript presented in an intelligible fashion and written in standard English?

Reviewer #1: Yes

Reviewer #2: Yes

5. Review Comments to the Author

Reviewer #1: Kyeongbin Baek et al. investigated the protective effects of live and UV-inactivated

influenza viruses via intraperitoneal and intramuscular administration. Cross-

protection against different subtypes of influenza virus was also investigated. The

compositions and administration routes of vaccination are important issue to

consider in order to enhance the efficacy of vaccines. In the case of the UV-

inactivated A/WSN/1993 (H1N1) virus, only a high dose of intraperitoneal

administration induced some degree of protection against intranasal challenge with

live A/WSN/1993 virus. However, low dose of live A/WSN/1993 virus via both routes

of administration showed complete protective effects. Intraperitoneal or

intramuscular administration with live- or UV-inactivated A/Philippines/2/1982 (H3N2)

virus produced no cross-protective antibodies, and resulted in no cross-protection

against intranasal challenge with A/WSN/1993. This information can be used as a

practical data in order to develop an effective influenza vaccine for cross-protection.

The experiments are well conducted with appropriate data analysis. Therefore, I

recommend the manuscript to be accepted for publication, while it is necessary to

finish the revision as follows:

1. The authors used ELISA to measure the amount of influenza A virus-specific

IgG. It will be necessary to describe whether the WSN or H3N2 Php virus

coated on 96-well plates is a live virus or a UV-inactivated virus.

2. In Figure 5, cross-reactive IgG was produced by intramuscular administration

of live viruses. However, cross-protection against different subtypes of

influenza virus was not observed. Could you add some more discussion with

the concept of threshold?

3. On line 316-318, there is a sentence ‘‘Additionally, the cytokine levels in sera

and lungs were not different than those in uninfected mice when the mice

were intranasally challenged with live WSN 14 days after intraperitoneal

inoculation with live H3N2 Php (data not shown).’’ Please show the results in

the supplementary data.

Reviewer #2: Here, Baek et al. reported a study on evaluating the impact of vaccination administration routes on the vaccine efficacy using both inactivated and live influenza viruses. They compared the Intramuscular and intraperitoneal inoculation. They assessed antibody responses (ELISA) and protective immune responses (viral lethal challenge study). Moreover, they tested the breath of immune responses induced by these routes of vaccination. They found that there was a protective immunity induced by either route of vaccination but only when a high does was used. Finally, they tested the breath of immune responses triggered by their immunization.

Overall, this manuscript targeted on an important topic, the impact of vaccination route on efficacy. The entire story will benefit from a more stringent experimental design, including critical information and more precise description.

Major points:

1. Given that they used IM and IP as their vaccination route, they induced humoral responses through systematic immune responses. Thus, cellular immunity should play an important role in regulating the humoral responses. To strengthen their conclusion, it is important to provide the audience some experimental information about cellular immune responses. Particularly, they tested the immune responses towards a heterologous virus. This is likely because of cellular responses towards the internal genes, which are more conserved between the viruses they tested. Therefore, it is important to present data about T cells responses after their vaccination.

2. In the manuscript, they evaluated the antibody by ELISA. This only gave the information about the level of specific antibodies, but not potency for those specific antibodies. HI or microneutralization assay should be adequate to address this concern.

3. The manuscript will benefit from a more precise description. For example, the live influenza virus used here is not a live attenuated vaccine. Another example would be the description of dosage used for challenge. Instead of using specific PFU, a much more meaningful way would use an equivalent MLD50 value. In such a way, the readers will be better informed about their challenge stringency.

Minor points:

1. Line 28, it is a typo for “Influenza A/WSN/1993”

2. There is a lack of consistency in experimental design. For example, in Figure 3, in this set of in vivo studies, they only observed animals 7 days post challenge. In their other in vivo studies, they observed animal 10 days post challenge.

3. The amount of protein (ug) would be a more proper unit for describing the dosage used for IM inoculation.

4. There is no indication to authenticate their inactivation.

6. PLOS authors have the option to publish the peer review history of their article (what does this mean?). If published, this will include your full peer review and any attached files.

Reviewer #1: No

Reviewer #2: No

---

## [Author Response · Author response to Decision Letter 0]

19 Sep 2022

We adjusted the format of our manuscript according to PLoS One 's style requirements. We indicated that “the funders had no role in study design, data collection and analysis, decision to publish, or preparation of the manuscript”. We also clarified the issue of data availability as follows: All relevant data are within the manuscript and its Supporting information file. We don’t have any other repository information for our data. We are sorry to make you confused. We inserted S1 Fig for the data not shown in the original manuscript. 

Point-to-point responses

Reviewer #1: Kyeongbin Baek et al. investigated the protective effects of live and UV-inactivated

influenza viruses via intraperitoneal and intramuscular administration. Cross-protection against different subtypes of influenza virus was also investigated. The compositions and administration routes of vaccination are important issue to consider in order to enhance the efficacy of vaccines. In the case of the UV-inactivated A/WSN/1993 (H1N1) virus, only a high dose of intraperitoneal administration induced some degree of protection against intranasal challenge with live A/WSN/1993 virus. However, low dose of live A/WSN/1993 virus via both routes of administration showed complete protective effects. Intraperitoneal or intramuscular administration with live- or UV-inactivated A/Philippines/2/1982 (H3N2) virus produced no cross-protective antibodies, and resulted in no cross-protection against intranasal challenge with A/WSN/1993. This information can be used as a practical data in order to develop an effective influenza vaccine for cross-protection. The experiments are well conducted with appropriate data analysis. Therefore, I recommend the manuscript to be accepted for publication, while it is necessary to finish the revision as follows:

1. The authors used ELISA to measure the amount of influenza A virus-specific IgG. It will be necessary to describe whether the WSN or H3N2 Php virus coated on 96-well plates is a live virus or a UV-inactivated virus.

Response : We used live viruses for ELISA to measure the amount of influenza A virus-specific IgG. Therefore, we described it in Line 174.

2. In Figure 5, cross-reactive IgG was produced by intramuscular administration of live viruses. However, cross-protection against different subtypes of influenza virus was not observed. Could you add some more discussion with the concept of threshold?

Response : As you pointed, the concept of threshold may explain our finding considering that the amounts of cross-reactive antibodies are small. In this revision, we additionally performed HI assays and found that the cross-reactive antibodies can’t neutralize another virus (Fig 2G and Fig 5G). Therefore, we became to know that the absence of cross-protection may be related with the functional properties of the specific antibodies induced by different subtypes rather than the quantity of the antibodies. Thanks. 

3. On line 316-318, there is a sentence ‘‘Additionally, the cytokine levels in sera and lungs were not different than those in uninfected mice when the mice were intranasally challenged with live WSN 14 days after intraperitoneal inoculation with live H3N2 Php (data not shown).’’ Please show the results in the supplementary data.

Response : According to your comments, the data were presented in Supporting information as S1 Fig.

Reviewer #2: Here, Baek et al. reported a study on evaluating the impact of vaccination administration routes on the vaccine efficacy using both inactivated and live influenza viruses. They compared the Intramuscular and intraperitoneal inoculation. They assessed antibody responses (ELISA) and protective immune responses (viral lethal challenge study). Moreover, they tested the breath of immune responses induced by these routes of vaccination. They found that there was a protective immunity induced by either route of vaccination but only when a high dose was used. Finally, they tested the breath of immune responses triggered by their immunization.

Overall, this manuscript targeted on an important topic, the impact of vaccination route on efficacy. The entire story will benefit from a more stringent experimental design, including critical information and more precise description.

Major points:

1. Given that they used IM and IP as their vaccination route, they induced humoral responses through systematic immune responses. Thus, cellular immunity should play an important role in regulating the humoral responses. To strengthen their conclusion, it is important to provide the audience some experimental information about cellular immune responses. Particularly, they tested the immune responses towards a heterologous virus. This is likely because of cellular responses towards the internal genes, which are more conserved between the viruses they tested. Therefore, it is important to present data about T cells responses after their vaccination.

Responses: 

As you commented, cellular responses are definitely important for vaccination effect. When we intranasally challenged with WSN after IP or IM inoculation with WSN or H3N2Php, the levels of virus-specific antibodies and their HI titers were significantly increased. Therefore, cellular immunity should be involved. However, to investigate the change of cellular immunity including T cell responses at this moment is difficult for us. Previously, we reported change of cell population including the increase of CD8+ cells as well as CD4+ T cells in response to IP injection with WSN (Frontiers in Immunology, 2019;10:1160. doi: 10.3389/fimmu.2019.01160). We wrote this finding in the introduction as follows in Line 67-74. 

“We previously demonstrated that intraperitoneal inoculation of mice with the Influenza H1N1 strain A/WSN/1933 (WSN) induced cross-reactive antibodies that facilitated heterosubtypic immunity to Influenza H3N2 strain A/Hongkong/4801/2014 [14,15]. Furthermore, we found that intraperitoneal inoculation with live influenza A virus altered immune cell populations at an early stage, resulting in depletion of B cells and macrophages along with immense neutrophil infiltration in the peritoneal cavity and bone marrow [15]. Expansion of the CD8+ T cell population in response to intraperitoneal inoculation with live influenza A virus likely played a role in cell-mediated protective immunity.”

In addition, we added discussion as follows in Line 377-381. Thanks for your thoughtful comments.

“In this study, we mainly revealed antibody production and compared outcome of the vaccinated mice after intranasal challenge. Considering that cellular immunity plays an important role in regulating the humoral responses and cellular responses toward the conserved genes of viruses may be critical for effective cross protection against heterologous viruses [4,24], further investigation on cellular immunity including T cell responses is required.”

2. In the manuscript, they evaluated the antibody by ELISA. This only gave the information about the level of specific antibodies, but not potency for those specific antibodies. HI or microneutralization assay should be adequate to address this concern.

Response : According to your comment, we performed HI assays and presented the data in Fig 2G and Fig 5G. We found that antibodies induced by one virus effectively inhibit hemagglutination mediated by the same virus but had no cross-protective activity to neutralize the other virus. As a result, we became to know that the absence of cross-protection may be related with the functional properties of the specific antibodies induced by different subtypes rather than the quantity of the antibodies. Thanks for your suggestion.

3. The manuscript will benefit from a more precise description. For example, the live influenza virus used here is not a live attenuated vaccine. Another example would be the description of dosage used for challenge. Instead of using specific PFU, a much more meaningful way would use an equivalent MLD50 value. In such a way, the readers will be better informed about their challenge stringency.

Response : We inserted the information regarding “live virus” in the materials and methods section, “Mice immunization and infection” as you pointed (Line 148-149). I adopted your expression as it is. Thanks.

 “The live influenza virus used here is not a live attenuated vaccine.”

We used 10 LD50 WSN virus for intranasal challenge as a lethal dosage. As your comment, we described the amounts of virus as the 10 LD50 WSN virus in Line 150-151. Thanks.

Minor points:

1. Line 28, it is a typo for “Influenza A/WSN/1993”

Response : We corrected as you pointed. Thanks.

2. There is a lack of consistency in experimental design. For example, in Figure 3, in this set of in vivo studies, they only observed animals 7 days post challenge. In their other in vivo studies, they observed animal 10 days post challenge.

Response : Actually, we observed the mice for 10 days in Figure 3 and 5. However, all the challenged mice died 7 days after challenge, therefore we made the graph up to 7 days. Considering your comment and to show the data more clearly, we changed the graph and showed the results up to 10 days. Thanks.

3. The amount of protein (ug) would be a more proper unit for describing the dosage used for IM inoculation.

Response : We used live or UV-inactivated virus solution for IM and IP inoculation. We used cell culture supernatants including viruses rather than purified virus, therefore we believe that pfu is more rational than the amount of protein (ug) for our experiments. Thanks for your consideration. 

4. There is no indication to authenticate their inactivation.

Response : We have indicated the authentication of the inactivation in the section “Ultraviolet inactivation of influenza A virus” of the material and methods as follows (line 114-117). “Inactivation of the viruses was confirmed by plaque assays showing no plaque-forming units after the UV exposure.” We marked the part for your convenience.

---

## [Editor Report · Decision Letter 1]

23 Sep 2022

Comparison of vaccination efficacy using live or ultraviolet-inactivated influenza viruses introduced by different routes in a mouse model

PONE-D-22-17990R1

Dear Dr. Kwon,

We’re pleased to inform you that your manuscript has been judged scientifically suitable for publication and will be formally accepted for publication once it meets all outstanding technical requirements.

Kind regards,

Juan Carlos de la Torre, Ph.D.

Academic Editor

PLOS ONE

Additional Editor Comments (optional):

In this revised version of their paper, the authors have adequately addressed most of the comments and concerns raised by the reviewers.

An issue that was not addressed relates to the T cell responses. However, the authors have provided a reasonable argument about the reasons why these data have not been incorporated into the revised version of their paper.
---

## [Editor Report · Acceptance letter]

30 Sep 2022

PONE-D-22-17990R1 

Comparison of vaccination efficacy using live or ultraviolet-inactivated influenza viruses introduced by different routes in a mouse model 

Dear Dr. Kwon:

I'm pleased to inform you that your manuscript has been deemed suitable for publication in PLOS ONE. Congratulations! Your manuscript is now with our production department. 

Kind regards, 

on behalf of

Dr. Juan Carlos de la Torre 

Academic Editor

PLOS ONE